# A human antibody against Zika virus crosslinks the E protein to prevent infection

S. Saif Hasan[1], Andrew Miller[1], Gopal Sapparapu[2,3], Estefania Fernandez[4], Thomas Klose[1], Feng Long[1], Andrei Fokine[1], Jason C. Porta[1], Wen Jiang[1,5], Michael S. Diamond[4,6,7,8], James E. Crowe Jr.[2,3,9], Richard J. Kuhn[1,5] & Michael G. Rossmann[1,5]

The recent Zika virus (ZIKV) epidemic has been linked to unusual and severe clinical manifestations including microcephaly in fetuses of infected pregnant women and Guillian-Barré syndrome in adults. Neutralizing antibodies present a possible therapeutic approach to prevent and control ZIKV infection. Here we present a 6.2 Å resolution three-dimensional cryo-electron microscopy (cryoEM) structure of an infectious ZIKV (strain H/PF/2013, French Polynesia) in complex with the Fab fragment of a highly therapeutic and neutralizing human monoclonal antibody, ZIKV-117. The antibody had been shown to prevent fetal infection and demise in mice. The structure shows that ZIKV-117 Fabs cross-link the monomers within the surface E glycoprotein dimers as well as between neighbouring dimers, thus preventing the reorganization of E protein monomers into fusogenic trimers in the acidic environment of endosomes.

[1] Department of Biological Sciences, Purdue University, West Lafayette, Indiana 47907, USA. [2] Department of Pediatrics, Vanderbilt University Medical Center, Nashville, Tennessee 37232, USA. [3] The Vanderbilt Vaccine Center, Vanderbilt University Medical Center, Nashville, Tennessee 37232, USA. [4] Department of Pathology & Immunology, Washington University School of Medicine, St Louis, Missouri 63110, USA. [5] Markey Center for Structural Biology and Purdue Institute for Inflammation, Immunology and Infectious Disease, Purdue University, West Lafayette, Indiana 47907, USA. [6] Department of Medicine, Washington University School of Medicine, St Louis, Missouri 63110, USA. [7] Department of Molecular Microbiology, Washington University School of Medicine, St Louis, Missouri 63110, USA. [8] Center for Human Immunology and Immunotherapy Programs, Washington University School of Medicine, St Louis, Missouri 63110, USA. [9] Department of Pathology, Microbiology and Immunology, Vanderbilt University, Nashville, Tennessee 37232, USA. Correspondence and requests for materials should be addressed to R.J.K. (email: kuhnr@purdue.edu) or to M.G.R. (email: mr@purdue.edu).

Zika virus (ZIKV) is a member of the *Flaviviridae* family of positive-stranded RNA viruses, which includes well-known human pathogens such as dengue (DENV), yellow fever (YFV), West Nile (WNV) and Japanese encephalitis (JEV) viruses[1]. ZIKV was first identified in sentinel rhesus monkeys in Uganda in 1947 (ref. 2), in mosquitoes in 1948 (ref. 2) and was first isolated from humans in 1952 (ref. 3). The first major outbreak of the virus was recorded in 2007 in Micronesia[4] and then in 2013–2014 in Oceania[5]. The latest outbreak, which started in Brazil in 2014–2015 (ref. 6), has spread to other countries in South America, North America and the Caribbean[7].

Flaviviruses contain an RNA genome ($\sim$11 kb) that is enclosed within the viral capsid. The nucleocapsid complex is enclosed within an icosahedral shell formed by 180 copies of each of the envelope ('E', $\sim$500 residues) and membrane ('M', $\sim$75 residues) glycoproteins. The M protein is a product of cleavage of the precursor M ('prM', $\sim$165 residues) protein. Cryo-electron microscopy (cryoEM) structures of mature ZIKV (refs 8,9) showed a typical 'herringbone' arrangement of the E proteins on the viral surface, which is a characteristic of mature flaviviruses[10]. Following cellular entry, flaviviruses assemble first as 'spiky', non-infectious immature particles with 60 trimeric spikes of E:prM heterodimers[11,12]. The immature particles undergo maturation into smooth infectious particles, which involves the cleavage of the prM protein into pr and M. The E and M glycoproteins are reorganized into 90 dimers[11,12]. Upon release into the extracellular environment, the pr peptide dissociates from the flavivirus surface to yield smooth, mature, infectious particles[11,13].

ZIKV infections were previously associated with only a rash and mild flu-like symptoms until the latest epidemic, during which a link was established between ZIKV infection and paralytic Guillian-Barré syndrome in adults[14], and fetal abnormalities in pregnant women, including microcephaly[15,16]. Four cases of fetal deformities have been reported in December 2016 in New York City[17]. In contrast to other flaviviruses that are spread mainly by arthropod vectors, recent evidence suggests that ZIKV can be transmitted sexually and vertically in addition to transmission by mosquitoes[18,19]. Given the severity of the symptoms caused by ZIKV infection in humans, it is crucial to understand the immune response elicited by ZIKV infection to develop neutralizing anti-ZIKV therapies. A recent study identified a human monoclonal antibody (mAb), ZIKV-117, that was demonstrated to possess therapeutic potential, with no cross-reactivity to other flaviviruses[20]. This is significant as ZIKV antibodies that cross-react with DENV have been shown to promote DENV infection by antibody dependent enhancement[21]. The mAb ZIKV-117 neutralizes ZIKV strains that belong to African, Asian and American lineages and is able to reduce fetal infection and death in mice. However, the mechanisms of neutralization of ZIKV infection by mAb ZIKV-117 and the structural basis for broad-range neutralization of ZIKV strains, while showing no cross-reactivity with other flaviviruses, had remained unknown.

Here we present a cryoEM map of ZIKV strain H/PF/2013 in complex with ZIKV-117 Fab at a resolution of 6.2 Å. The map showed that the mechanism of virus neutralization was by binding of Fab fragments that cross-linked the E monomers within dimers and also cross-linked neighbouring dimers in the viral glycoprotein shell. However, the binding of any one of the Fab molecules excluded binding of a Fab to any of the other chemically equivalent sites in the icosahedral asymmetric unit. Thus, saturation is achieved by binding of only 60 Fabs to the 180 chemically equivalent sites on the virus, greatly reducing the concentration of the antibody needed for neutralization.

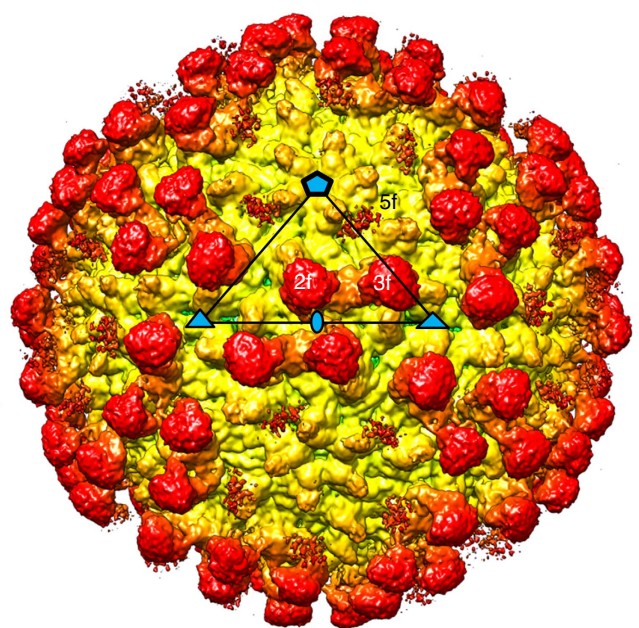

**Figure 1 | CryoEM map of ZIKV/ZIKV-117.** The ZIKV-Fab surface is shown in radial colouring, from yellow ($\sim$210 Å) to red ($\sim$290 Å). The bound Fab (red 'knobs') extends from the surface of ZIKV, which is shown in yellow. The Fab sites close to the two-fold ('2f'), three-fold ('3f') and five-fold ('5f') axes are labelled. The icosahedral 2 (oval), 3 (triangle) and 5 (pentagon) fold axes are marked in blue shapes with black outlines, and a representative asymmetric unit is shown as a black triangle.

## Results

**The cryoEM structure of ZIKV complexed with ZIKV-117 Fab.** Mature ZIKV was purified from Vero cells over-expressing the enzyme furin[8]. MAb ZIKV-117 was isolated from a B cell in a peripheral blood sample, obtained from an otherwise healthy human subject with previous history of symptomatic ZIKV infection[20]. The ZIKV particles were incubated with Fab fragments of ZIKV-117, flash-frozen on lacey carbon EM grids and imaged using a Gatan K2 direct electron detector attached to a Titan-Krios microscope. After automated selection of virus-Fab particles, a data set containing 8,153 virus-Fab complexes was split into two halves to perform two independent 3D-reconstructions for a 'gold-standard' evaluation of the resolution[22]. The previously published ZIKV cryoEM map[8] was low-pass filtered to 50 Å resolution, and used as the initial model for single particle reconstruction by the program *jspr* (ref. 23). The final map had a resolution of 6.2 Å corresponding to a Fourier Shell Correlation coefficient of 0.143 (ref. 24) (Supplementary Fig. 1). The cryoEM map (Fig. 1) was interpreted by fitting of E protein and Fab coordinates with the programs Chimera[25], Coot[26] and EMfit[27] following the original procedure first used to determine the structure of DENV[10]. The icosahedral asymmetric unit of the ZIKV structure contains three copies of the E protein, whose ectodomain is organized into three domains, DI, DII and DIII (ref. 28) (Supplementary Fig. 2a,b). The structure of a ZIKV E dimer (Protein Data Bank, PDB ID 5IRE, residues 1–396) (ref. 8) was fitted into the cryoEM map by superposing the dimer two-fold axis onto an icosahedral two-fold axis ('i2 axis') (Fig. 2, monomers A and A') using the Chimera and EMfit programs. This dimer will be referred to as the icosahedral E dimer ('i2 dimer'). A second E dimer (Fig. 2, monomers C and E) was fitted at a general position, where the cryoEM map appears to have a quasi two-fold axis ('q2 axis', Fig. 2, yellow oval). This dimer will be referred to as the general

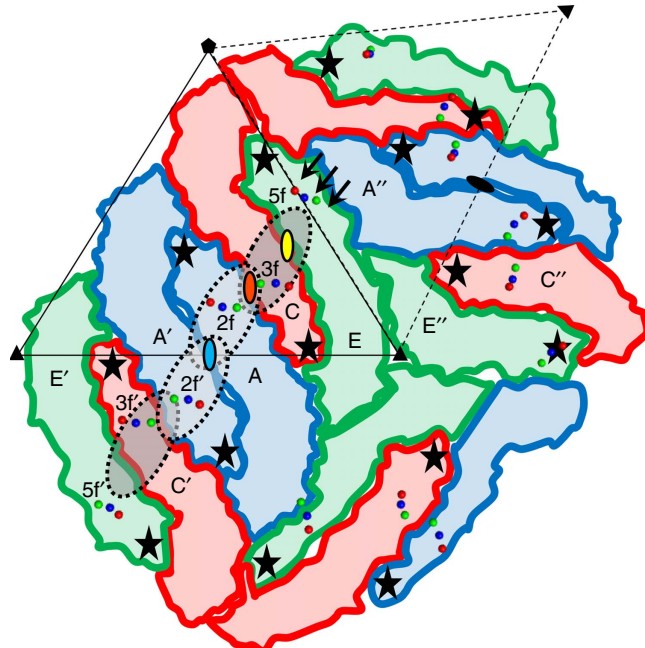

**Figure 2 | Fab binding sites on ZIKV surface.** The E proteins are represented in three colours—blue (chains A, A' and A''), red (chains C, C' and C'') and green (chains E, E' and E''). Chains A and A' are arranged in a dimer around the icosahedral two-fold axis (blue oval with black outline), while chains C and E are arranged in a general dimer around a quasi two-fold axis (yellow oval with black outline). The icosahedral and general dimers are related by an axis of quasi two-fold symmetry (orange oval with black outline). Residues Asp67 (red), Gln89 (green) and Lys118 (blue), which were mapped to the Fab binding site by mutagenesis, are shown as spheres. The footprints of the Fab bound at the 2f and 3f sites are represented as light and dark grey ovals with broken outlines. The black stars mark the position of the fusion peptides. The steric hindrance on the 5f site due to the orientation of DIII of chain A'' is highlighted by black arrows. For reference, two asymmetric units are shown as triangles in solid and broken dashed lines. The icosahedral five- and three-fold axes are marked with a pentagon and two triangles receptively.

E dimer ('q2 dimer'). As the asymmetric unit contains three copies of the E protein (Fig. 2, monomers A, C and E), it should be possible for three Fabs to be bound within the asymmetric unit. However, only two prominent Fab-bound positions were observed, one ('2f') closest to an i2 axis and the other ('3f') closest to an icosahedral three-fold axis ('i3 axis') (Figs 1 and 2). The additional density at the third chemically equivalent position ('5f') closest to an icosahedral five-fold axis ('i5 axis') was weak (Fig. 1). The coordinates of a ZIKV-specific Fab (ZV-67, PDB ID 5KVG) (ref. 29) were split into the constant and variable domains of the heavy and light chains. These domains were fitted individually into the Fab densities at the 2f and 3f sites using Chimera and Coot. As the density is incomplete at the 5f site, Fab coordinates were placed by hand. The average heights of electron density of the E proteins and the Fabs were compared using the program EMfit.

**Interference between overlapping ZIKV-117 Fab binding sites.** Within an icosahedral asymmetric unit of ZIKV, the two E protein monomers of the general dimer (Fig. 2, monomers C and E) are related by q2 symmetry (Fig. 2, yellow oval). The icosahedral E dimer (Fig. 2, monomers A and A') is similar in structure to the general E dimer (Fig. 2, monomers C and E). As a result, the

icosahedral i2 dimer and the general q2 dimer are related by a q2 axis (Fig. 2, orange oval). Three residues of the ZIKV E protein, Asp67, Gln89 and Lys118 ('Asp-Gln-Lys triad'), previously determined to be in the ZIKV-117 epitope by alanine mapping[20], are located closest to the q2 axis (Fig. 2, orange oval) that relates the i2 dimer with the general q2 dimer. The similar positions of these three residues in the footprint of the Fabs at both the 2f and 3f sites are satisfactorily consistent with the EM results presented here and also demonstrate the chemical equivalence of these sites.

Binding of Fabs at the 5f sites around each five-fold axis is not limited by overlap between Fabs bound to neighbouring 5f sites because these sites are spatially well separated (Supplementary Fig. 3). However, the low occupancy of Fabs at the supposedly chemically equivalent 5f sites is readily understandable because of the different environments of the 5f sites to the 3f and 2f sites. The 'road map' (Fig. 3) shows that the 5f site is bounded at one side by an E monomer that extends radially outwards further than the 5f site. Therefore, the E monomer not only changes a part of the 5f site to be different to the 3f and 2f sites, but also sterically hinders the binding of Fabs to the 5f sites. The residual small occupancy of the 5f sites (Fig. 1) is probably due to the inclusion of particles in the cryoEM reconstruction that were able to bind Fab at the 5f sites after inducing small local conformational changes.

The footprints of Fabs when bound to the 3f and 2f sites overlap (Figs 2 and 3). This implies that binding of Fabs at these sites within an icosahedral asymmetric unit is mutually exclusive. Only one Fab can be bound within an asymmetric unit, either at the 2f or the 3f site. Furthermore, the footprint of the 2f site overlaps the footprint of the two-fold related 2f' site (Figs 2 and 3). Thus, similarly only one of the 2f or 2f' sites can be occupied at a time. Lastly the binding of Fab to the 2f' and 3f' sites is also mutually exclusive. Assuming that each binding site is chemically identical and therefore equally likely to bind a Fab molecule, there are three different possible arrangements of Fabs that can bind to the overlapping 3f, 2f, 2f' and 3f' sites (Supplementary Fig. 4). Inspection of Supplementary Fig. 4 shows that on average, the 3f and 3f' sites will be occupied two times out of every three randomly chosen particles, whereas the 2f and 2f' sites will be occupied only once out of three randomly chosen particles. Thus, on average, the 3f site should have 67% occupancy whereas the 2f site should have 33% occupancy.

The Fab occupancy at the 2f and 3f sites relative to the ZIKV E protein was determined by the program EMfit, which calculates the average electron density at the atomic centres of the fitted coordinates. The average heights of density of the Fabs bound at the 2f and 3f sites were 35 and 58% respectively of the E protein density (100% occupancy, Supplementary Table 1), which correspond to 21 and 35 Fabs at the 2f and 3f sites. The missing 7% of Fabs is either because the average of all the particles used in the EM reconstruction had less than 60 Fabs per particle or is simply experimental error. Given the experimental conditions, the observed occupancies are in excellent agreement with the predicted 33 and 67% (Supplementary Fig. 4).

## Discussion

The efficacy of neutralization of ZIKV infection by ZIKV-117 is in part due to the requirement of only 60 bound Fab molecules instead of the anticipated stoichiometric 180 bound Fabs to effectively cross-link the glycoprotein shell. In addition, the tight binding of the Fab to the E protein is also a significant contributing factor. The footprints of the Fabs bound at the 2f and 3f sites consist primarily of polar residues of the E protein (Fig. 3), which can participate in the formation of salt bridges and

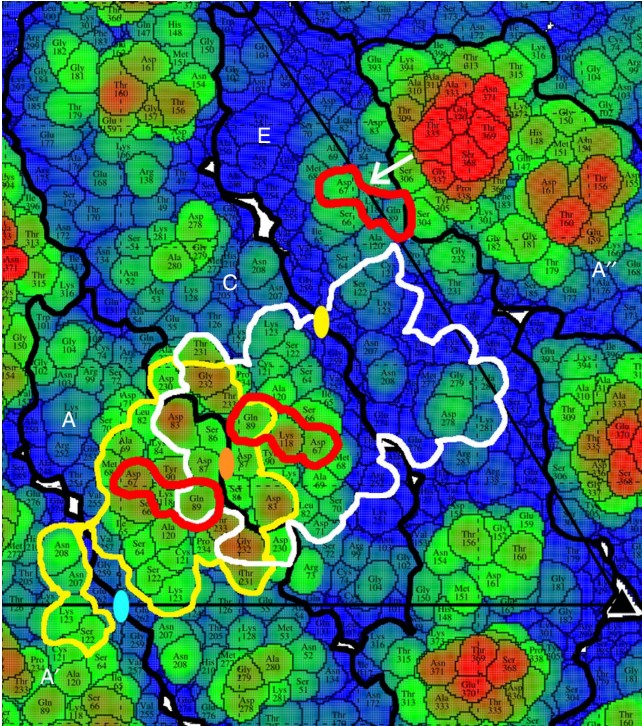

**Figure 3 | Footprints of ZIKV-117 Fabs on the E protein.** The figure shows a roadmap of the ZIKV E protein. The E protein residues are coloured according to their distance from the virus centre, with the residues closer to the centre coloured blue (218 Å) and those farthest from the centre coloured red (235 Å). The colours of residues transition from blue to red through green. The position of the icosahedral two-fold axis is marked with a blue oval. The quasi two-fold axes are marked as orange or yellow ovals. The outlines of the individual E protein chains are marked with thick black lines, and the chains are labelled as per the convention described in Fig. 2. The residues Asp67, Gln89 and Lys118 within the ZIKV-117 epitope are highlighted with red lines. The ZIKV-117 footprints formed by E protein residues that lie within 8 Å of the bound Fab at the 2f (yellow outline) and 3f (white outline) sites are highlighted across the quasi two-fold axis (orange oval). The footprints extend across multiple E proteins, and there is significant overlap between the footprints. At the 5f site, DIII of chain A″ (highlighted with a white arrow) poses a steric barrier to the binding of Fab.

hydrogen bonds with the ZIKV-117 Fab to form strong, stable virus-Fab interactions. In addition, the Fab footprints show that ZIKV-117 cross-links E monomers within a dimer, and across neighbouring dimers (Fig. 3). At the 2f site, the Fab cross-links monomers A and A′ of the i2 dimer and monomer C of the general q2 dimer (Figs 2 and 3). In contrast, the Fab bound at the 3f site cross-links monomers C and E of the general q2 dimer and monomer A of the i2 dimer (Figs 2 and 3). Therefore, ZIKV-117 causes extensive intra-dimer as well as inter-dimer cross-linking of the ZIKV E protein.

Antibodies that cross-link E proteins and neutralize infection have been reported for ZIKV[30], DENV[31–35] and WNV[36]. These antibodies cross-link DII with either DI or DIII, or with both DI and DIII. The ZIKV-specific antibody binds to DIII at 120 of 180 sites, away from the E protein dimer axis[30]. A structure of a DENV antibody bound to ZIKV shows that the antibody binds at 180 sites on the virus particle to cross-link DI, DII and DIII to prevent low pH triggered conformational changes of the virus[37]. The epitopes of the cross-linking ZIKV, DENV and WNV antibodies are not close to the i2 or q2 axes. Therefore, in

contrast to ZIKV-117, the binding of Fab to one site does not exclude binding of Fab to another site. This implies that antibodies that bind to sites on DII are likely to require a lower concentration to saturate and neutralize the virus.

The flavivirus E dimer is converted into a trimer upon viral entry into host cells under acidic conditions in endosomes[38–40]. These trimers expose the fusion peptides which insert themselves into the endosomal membrane to form a channel across the endosomal membrane for the transfer of the viral RNA genome into the host's cytosol[41]. Binding of the ZIKV-117 Fab at the 2f and 3f positions will inhibit the low-pH triggered rearrangement of the E protein dimer into a fusogenic trimer inside the host cell. This is supported by a pre-/post-neutralization assay of ZIKV infection, which shows that the ZIKV-117 is able to significantly inhibit infection at a post-attachment, intra-cellular step (Supplementary Fig. 5).

At acidic pH mature flaviviruses undergo a major conformational change in which the E glycoproteins re-arrange themselves to form 60 trimeric spikes[42] with the fusion peptide in DII of each monomer exposed at their distal extremity. This large re-arrangement is likely to leave the virus lacking homogeneity accounting for the decline in obtaining high resolution cryoEM structures[37]. Inspection of Fig. 2 suggests various ways in which the monomers might re-arrange themselves to extend radially with their fusion peptides at the far end of the spikes away from the virus centre. For instance, monomers A, C and E (Fig. 2) are close together to form a spike, and monomers A′, C′ and E′ would then be able to form a two-fold related spike. This would allow the formation of one spike per icosahedral asymmetric unit. However if monomers A′, A and C were to form a spike, they would leave monomers E, C′, E′ without partners. This would allow the formation of one spike for two icosahedral asymmetric units. But acidification of DENV results in the formation of 60 spikes per particle, which is one spike per asymmetric unit[41]. Therefore, the first option mentioned here is a more likely association of monomers that make a fusogenic virus. As the overlap of ZIKV-117 Fab binding sites especially emphasizes the cross linking of monomers A, C and E, it is probable that the antibody will inhibit the formation of fusogenic spikes. This also shows that the binding of 60 Fabs is enough to inhibit 90 E dimers from making a spikey fusogenic structure.

Since the beginning of the 2013 ZIKV outbreak, emphasis has been placed on the study of cross-reacting anti-DENV antibodies for neutralizing ZIKV infection[21,37,43–46]. In vivo studies have shown that anti-ZIKV antibodies that cross-react with DENV may promote infection by antibody-dependent enhancement[21]. The ZIKV-117 mAb does not bind any of the four DENV serotypes or WNV[20]. An alignment of flavivirus sequences shows that the E protein residues that form the ZIKV-117 footprint are poorly conserved across flaviviruses (Supplementary Fig. 6a). In contrast, the footprint of ZIKV-117 is largely conserved among ZIKV strains (Supplementary Fig. 6b). Consistent with recent functional data showing potent neutralization of five different African, Asian and American strains of ZIKV, the ZIKV-117 mAb binds to a conserved ZIKV epitope and should therefore provide broad-range protection against virtually all strains of ZIKV.

In summary, considering that the surface of ZIKV is made of 60 copies of three environmentally different E monomers, it would be expected that 180 copies of the Fab would be bound to the ZIKV virion. Our study shows that contrary to expectation, Fab binding at the 5f site is extremely weak despite no obvious clashes between symmetry related Fabs around the five-fold axes. We show that the epitope at the 5f site has limited accessibility due to steric hindrance from a neighbouring E monomer. Furthermore, our study shows that the overlap between Fab

binding sites limits the occupancies of the 3f and 2f sites in a predictable manner, implying that neutralization of the virus can be achieved with a lower occupancy of antibodies compared to what would be the case were the binding sites well separated. The structure of the Fab-virus complex suggests that the antibody functions by preventing the rearrangement of E proteins necessary for low-pH mediated fusion.

## Methods

**Isolation of ZIKV-117 Fab fragments.** MAb ZIKV-117 is a naturally occurring fully human mAb isolated from a B cell in a peripheral blood sample obtained from an otherwise healthy subject with previous history of symptomatic ZIKV infection. The isolation and determination of the variable region sequences of the heavy and light chains were described previously[20]. RT-PCR was performed with primers having overlapping regions with the leader and constant domains[47], and the amplicons were cloned directly into immunoglobulin expression vector cassettes[48] using a Gibson assembly kit (New England Biolabs). Plasmids encoding the heavy or light chains were transfected transiently into Expi293 cells (Invitrogen), and the secreted Fab protein was harvested from culture supernatant. Recombinant Fab protein was concentrated and purified by affinity chromatography using the CaptureSelect IgG-CH1 matrix (Thermo Scientific) and further concentrated using 10 MWCO concentrators (Millipore).

**CryoEM and 3D reconstruction.** The ZIKV particles were incubated with the ZIKV-117 Fab at 4 °C for 2–3 h, and then flash-frozen (2.5 μl) on lacey carbon EM grids by plunging in liquid ethane using a Gatan Cp3 plunger. Micrographs were recorded on a Gatan K2 direct electron detector, attached to a Titan-Krios microscope at a magnification of × 18,000 in the direct counting mode with a pixel size of 1.62 Å. The total electron dose per image was ∼30 e Å$^{-2}$. The automated software Leginon[49] was used to collect 1,141 micrographs, of which 406 micrographs were rejected due to poor ice thickness, drift and errors in defocus. A total of 38,835 particles of the Fab-virus complex were selected in Appion[50] from the remaining 735 micrographs, within a defocus range of −2.5 to −5.0 μm. The 38,853 particle set was then subjected to two rounds of non-referenced 2D-classification in RELION[51] to generate a subset of 8,153 homogeneous Fab-virus particles, which was split into two half data sets to perform two independent 3D-reconstructions for a 'gold-standard' evaluation of the resolution[22]. The previously published ZIKV cryoEM map[8] was low-pass filtered to 50 Å resolution, and used as the initial model for single particle reconstruction by the program jspr (ref. 23). During the initial stages of the reconstruction, the centre position and orientation of each particle were refined using projection matching, which was followed by the refinement of high-order parameters, that is, anisotropic magnification, astigmatism, defocus and beam tilt[52,53]. A resolution of 6.2 Å was assigned corresponding to a Fourier Shell Correlation coefficient of 0.143 (Supplementary Fig. 1). The fitting procedure for the E protein and Fab has been described in the main text.

**Pre- and post-attachment neutralization assays.** For the pre-attachment assay, serial dilutions of recombinant ZIKV-117 were pre-incubated with 10$^2$ FFU of ZIKV Brazil Paraiba 2015 for 1 h at 4 °C. The mAb-virus complexes then were added to Vero cell monolayers in 96-well plates for 1 h at 4 °C. For the post-attachment assay, 10$^2$ FFU of ZIKV Brazil Paraiba 2015 was added to a Vero cell monolayer in 96-well plates for 1 h at 4 °C. Unbound virus was removed followed by serial washes with Dulbecco's modified Eagle medium (DMEM), and then dilutions of recombinant ZIKV-117 were added for 1 h at 4 °C. Subsequently, all cells were washed three times with DMEM, incubated in DMEM supplemented with 5% FBS at for 1 h at 37 °C, and then overlaid with 1% (w/v) methylcellulose in α-MEM supplemented with 4% heat-inactivated FBS. Plates were fixed 40 h later with 1% paraformaldehyde in PBS for 1 h at room temperature. The plates were incubated sequentially with 500 ng ml$^{-1}$ mouse anti-ZIKV (ZV-16, E. Fernandez and M. Diamond, unpublished) and HRP-conjugated goat anti-mouse IgG in PBS supplemented with 0.1% (w/v) saponin (Sigma) and 0.1% BSA. ZIKV-infected cell foci were visualized using TrueBlue peroxidase substrate (KPL) and quantitated on an ImmunoSpot 5.0.37 macroanalyzer (Cellular Technologies).

**Sequence alignments.** The sequences of flavivirus ZIKV E protein were obtained from GenBank. Alignment of E protein sequences was performed using the program Clustal-Omega[54] and figures were prepared using the program ESprit[55].

**Data availability.** The cryoEM map and fitted coordinates of the ZIKV/ZIKV-117 complex have been deposited in the Electron Microscopy Data Bank and the Protein Data Bank under the accession codes EMD-8548 and 5UHY respectively. The data that support the findings of this study are available from the corresponding author on request. Requests for antibody material should be addressed to james.crowe@vanderbilt.edu.

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

## Acknowledgements

We thank Yinguan Sun and Valorie D. Bowman (Purdue University) for technical help in cryoEM data collection, Yue Liu and Vidya Mangala Prasad (Purdue University) for advice in fitting of coordinates, and Chuan Xiao (University of Texas at El Paso) for assistance in drawing roadmaps. CryoEM data were collected at the Purdue University CryoEM facility. This work was supported by U.S. National Institutes of Health grant awards R01 AI076331 (to M.G.R.), R01 AI073755 (to M.S.D., sub-awards to R.J.K., M.G.R. and J.E.C.) and R01 AI104972 (to M.S.D.) as well as National Institutes of Health contract HHSN272201400024C (to J.E.C.).

## Author contributions

S.S.H., A.M., G.S., T.K., F.L. and E.F. performed the experiments. S.S.H., T.K., J.C.P., A.F., E.F., M.S.D., J.E.C., R.J.K. and M.G.R. performed data analysis. W.J. made *jspr* available for reconstruction and refinement of cryoEM map. S.S.H., R.J.K. and M.G.R. wrote the initial draft of the manuscript with all authors contributing to editing into the final form.

## Additional information

**Competing financial interests:** M.S.D. is a consultant for Inbios and Visterra, on the Scientific Advisory Boards of Moderna and OraGene, and a recipient of grants from Moderna and Visterra. J.E.C. is a consultant for Sanofi and Ridgeback Biotherapeutics, is on the Scientific Advisory Boards of PaxVax, CompuVax, GigaGen, Meissa Vaccines and is a recipient of research grants from Moderna and Sanofi. The remaining authors declare no competing financial interests.

**Publisher's note**: 

