## [Peer Review File · Nature Communications]

28

29 **Supplementary Figure 1 | Fourier shell correlation (FSC) curve as a function of**
30 **resolution⁻¹.** The curve was calculated between two independent reconstructions of the ZIKV-
31 Fab complexes. The resolution that corresponds to an FSC coefficient of 0.143 was 6.2 Å.

32

33

34

35

36

37

38

39

Supplementary Figure 2 | The ZIKV E protein ectodomain. **a.** The ectodomain of the E monomer is divided into three domains, DI (red, residues 1-50, 133-195, 284-301), DII (yellow, residues 51-132, 196-283), and DIII (blue, residues 302-396). **b.** The E monomer is arranged as a symmetric dimer across an axis of 2-fold symmetry (blue oval with black outline).

40
41
42
43
44
45
46
47
48
49

Supplementary Figure 3 | Binding of Fab at the 5f site is not hindered by steric clashes. The figure shows a surface representation of the ZIKV/ZIKV-117 cryoEM map oriented down an icosahedral 5-fold axis. The Fabs bound at the 2f, 3f and 5f sites are colored in magenta, blue and green respectively. Binding of Fabs to the 5f sites related by icosahedral 5-fold symmetry does not lead to steric hindrance between symmetry related Fabs. For reference, an asymmetric unit is shown as a triangle with the axes of 5 (pentagon), 3 (triangle), and 2 (oval) fold symmetry shown in blue shapes with black outlines.

50

3f	2f	2f'	3f'
✓	X	✓	X
✓	X	X	✓
X	✓	X	✓
2	1	1	2

51

52

53 **Supplementary Figure 4 | Probability of ZIKV-117 Fab binding at 2f and 3f sites.**

54 Due to the close proximity between the 2f and 3f sites within an asymmetric unit and of related

55 2f' and 3f' sites in an adjacent asymmetric unit, simultaneous binding to all sites is not possible.

56 The figure shows a schematic outline of all possible combinations of simultaneously bound

57 Fabs. Two adjacent asymmetric units are represented as triangles. The icosahedral 5

58 (pentagon), 3 (triangle), and 2 (oval) fold axes are shown in blue shapes with black outlines.

59 The 2f, 3f, 2f', and 3f' binding sites for Fab are shown as ovals. An empty oval represents an

60 unoccupied site, while a solid black oval indicates a bound Fab. The table summarizes the three

61 states that have been described schematically. The check mark represents the presence of a

62 Fab while a cross indicates the absence of a Fab. The 3f site has twice the probability of being

63 occupied by a Fab than the 2f site.

64

65

ZIKV-117 inhibitor

66

67

68

69

70

71

72

73

Supplementary Figure 5 | Neutralizing activity of ZIKV-117 post-attachment. The figure shows the pre- (black trace) and post-attachment (red trace) neutralization of ZIKV infection by ZIKV-117. ZIKV-117 is able to neutralize ZIKV infection after attachment of ZIKV to host cells, which indicates that ZIKV-117 potentially inhibits an intracellular step in ZIKV infection.

74
75
76
77
78
79
80
81
82
83
84
85
86
87
88
89
90

Supplementary Figure 6 | Conservation of the ZIKV-117 footprint among flaviviruses. An alignment of E protein residues that form the 2f and 3f footprints are shown for common pathogenic flaviviruses are shown in **panel a**, and for ZIKV strains in **panel b**. Completely conserved positions are highlighted in red, partially conserved in yellow, and the positions that lack any conservation have a white background. Residues Asp67, Gln89 and Lys118 are highlighted with arrows. **a.** The footprints of ZIKV-117 Fab bound at the 2f and 3f sites are poorly conserved between flaviviruses. **b.** In contrast, the footprints are well-conserved in the ZIKV strains across the strains of Asian (Malaysia P6740 and H/PF/2013), African (Dakar and MR-766) and American (Paraiba and SPH2015) lineage. GenBank accession numbers for sequences shown in **panel a**: ZIKV_H/PF/2013, KJ776791; ZIKV_Mr766, AY632535; DENV1, 07K3640DK; DENV2, NC_001474; DENV3, EU081190; DENV4, GQ398256; WNV, DQ211652; JEV, D90194; YFV, AY640589. GenBank accession numbers for sequences shown in **panel b** (ZIKV strains): Malaysia, HQ234499.1; H/PF/2013, KJ776791; Dakar, HQ234501.1; MR-766, KU720415.1; Paraiba, KX280026.1; SPH2015, KU321639.1.

91

Supplementary Table 1 | Occupancy of Fab estimated by EMfit

	Sumf (Cα)	Occupancy (%)	Copies
E dimer (icosahedral, chains A-A')	38.5	100	30
E dimer (general, chains C-E)	37.8	100	60
Fab (2f)	13.5	35	21
Fab (3f)	22.0	58	35

92

93

94

95

96

97

98

Sumf is the average density around the atoms of a fitted domain. Only the C α atoms were included in the calculation. Occupancy of each Fab was calculated relative to the average E protein sumf value, *i.e.*, 38.2. For location of the E dimers in the cryoEM map and nomenclature of the chains, see Fig. 2 in the main text.